# Nonlinear dynamic wave properties of travelling wave solutions in in (3+1)-dimensional *mKdV−ZK* model

S. M. Yiasir Arafat⬤*, M. Asif, M. M. Rahman

Department of Mathematics, Bangladesh University of Engineering and Technology, Dhaka, Bangladesh

* yiasirarafat28@gmail.com

## Abstract

The (3+1)-dimensional *mKdV−ZK* model is an important framework for studying the dynamic behavior of waves in mathematical physics. The goal of this study is to look into more generic travelling wave solutions (TWSs) for the generalized ion-acoustic scenario in three dimensions. These solutions exhibit a combination of rational, trigonometric, hyperbolic, and exponential solutions that are concurrently generated by the new auxiliary equation and the unified techniques. We created numerous soliton solutions, including kink-shaped soliton solutions, anti-kink-shaped solutions, bell-shaped soliton solutions, periodic solutions, and solitary soliton solutions, for various values of the free parameters in the produced solutions. The attained solutions are displayed geometrically in the surface plot (3-D), contour, and combined two-dimensional (2-D) figures. The combined 2-D figure would make it easier to understand the impact of the speed of the wave. Based on time, the influence of the nonlinear parameter *β* on wave type is comprehensively investigated using various figures, demonstrating the significant impact of nonlinearity. These graphical representations are based on specific parameter settings, which help to grasp the model's intricate general behavior. However, the results of this research are compared with the outcomes obtained in published literature executed by other scholars. The results indicate the approach's effectiveness and reliability, making it suitable for widespread use in a range of sophisticated nonlinear models. These techniques successfully generate inventive soliton solutions for various nonlinear models, which are crucial in mathematical physics.

**Data Availability Statement:** The data are all contained within the manuscript.

**Funding:** The author(s) received no specific funding for this work.

## 1 Introduction

In a wide range of scientific, technical, and technological domains, theoretical outcomes are superior to experimental results in comprehending physical processes. Real characteristic phenomena of our world are described with the help of nonlinear evolution models (NLEMs). There are a lot of applications of NLEMs in several domains of physics, mathematics, and engineering. For instance, plasma physics [1,2], fluid dynamics [3,4], biomathematics [5], nonlinear optics [6,7], shallow water waves [8,9], and many others [10–15]. Many researchers also discussed phase portraits, maximal Lyapunov exponents, Neimark–Sacker bifurcation, period-

**Competing interests:** The authors have declared that no competing interests exist.

doubling bifurcation diagrams, stability analysis, exploring fractional order and many other physical phenomena [16–27]. It is a very challenging task to find the exact solution for NLEMs, but researchers are very interested in discovering potent and effective techniques. Because of the widespread applications and importance of nonlinear models, many scholars have developed a variety of useful tools and approaches to analysis partial differential equations (PDEs) such as, $\left(\frac{G\prime}{G}\right)$-expansion approach [28,29], F-expansion method [30], tanh-coth method [31], first integral technique [32], direct algebraic method [33], inverse scattering method [34], sin-Gordon expansion [35], new auxiliary equation approach [36], modified simple equation method [37], Hirota bilinear method [38,39], unified method [40], modified modified exp-function method [41–43] and so on.

The (3+1)-dimensional *mKdV−ZK* model describes the weakly nonlinear ion-acoustic waves in a magnetized electron-positron plasma with equally hot and cool components [44]. In addition, it describes the evolution of ion-acoustic disturbances in a magnetized plasma with two negative ion components of different temperatures. Therefore, the (3+1)-dimensional *mKdV−ZK* model provides a base for theoretical research and application across a wide range of scientific areas. The (3+1)-dimensional *mKdV−ZK* [45] model is given in the following form:

$$u_t + \beta u^2 u_x + u_{xxx} + u_{xyy} + u_{xzz} = 0, \tag{1.1}$$

where $\beta$ is an arbitrary constant that measures the nonlinearity's strength, $u$ is a wave function with $x,y,z$, and $t$ are the independent variables, the subscripts stipulate the partial derivatives. Also, $u^2 u_x$ is a nonlinear term known as cubic nonlinear that indicates the wave profile effect on linear wave propagation and represents cubic relationships.

Throughout history, numerous efficient and compatible methodologies have been used to analyze the *mKdV−ZK* model such as, the improved fractional sub-equation method [46], the modified Riemann–Liouville derivative, the exp-function, the $\left(\frac{G\prime}{G}\right)$-expansion, and the generalized Kudryashov method [47], the improved generalized Riccati equation mapping method [48], the improved Bernoulli sub-equation method [49], the Sardar-sub equation method [50]. the variable separated ODE method [51] and more.

Several researchers have investigated the new auxiliary equation and unified methodologies to examine the exact TWSs of a large number of NLEMs. To the best of our knowledge, researcher has not been found TWSs to the (3+1)-dimensional *mKdV−ZK* model through new auxiliary equation and unified methods.

Therefore, in the above-mentioned academic study, the core determination of our article is to provide standard, compatible, and stable TWSs to the specified model using these methods. Moreover, we represent the physical illustration as well as the 3-dimensional, contour and combined 2- dimensional graphical depiction are displayed to the investigated solutions. Understanding the (3+1)-dimensional mKdV-ZK model is theoretical knowledge; its understanding of wave behavior also enables scientists and engineers to create more efficient communication networks and better understand wave interactions in natural environments. The outcomes of this study will be significant in clarifying the meaning of the difficult physical phenomena in fluid mechanics, high-frequency plasma physics, marine engineering, ocean physics, and solitary wave theory.

The rest of the manuscript is organized as follows: In part 2, we discuss the methodology including new auxiliary equation unified methods as well as discuss the application of the stated model via abovementioned methods in part 3. The outcomes of these model are graphically present and give physical explanation with real life significant in part 4. Finally, the paper's conclusion is reached.

## 2.Overview the schemes

We will briefly describe two analytical techniques for exploring some soliton solutions of the PDEs in this section. We suppose the PDEs with variables *z,y,z*, and *t* of a function *F* in the desired construction:

$$F(u, u_x, u_z, u_y, u_t, u_{xx}, u_{xy}, u_{xz}, u_{xt}, u_{yx}, u_{yy}, u_{yz}, u_{yt}, u_{zz}, u_{tt}, \ldots) = 0, \tag{2.1}$$

in this case, *F* is a nonlinear polynomial function covering wave function *u(x,y,z,t)*, including its various partial derivatives respect to *x,y,z*, and *t*. By using a proper wave transform

$$u(x, y, z, t) = u(\xi), \quad \xi = px + qy + rz - \omega t, \tag{2.2}$$

Eq (2.1) becomes

$$H(u, u', u'', u''', \ldots) = 0. \tag{2.3}$$

The symbol (′) indicates the derivative with respect to $\xi$.

### 2.1 The new auxiliary equation scheme

In this sub-section, we describe the total procedure to the new auxiliary equation method [36]. We assume that the TWSs to the Eq (2.3) are the following form:

$$u(\xi) = \sum_{j=0}^{S} k_j a^{jg(\xi)}, \tag{2.4}$$

where $k_j(j = 0,1,2,\ldots,S)$ are constants to be compute, such that $k_j \neq 0$ and $g(\xi)$ satisfies the following equation

$$g'(\xi) = \frac{1}{\ln(a)} \left\{ ma^{-g(\xi)} + n + la^{g(\xi)} \right\}, \tag{2.5}$$

we determine the positive integer *S* applying the balancing between the highest order derivatives and highest order nonlinear terms in Eq (2.3). Substituting Eqs (2.4) and (2.5) into Eq (2.3), producing an algebraic equation where the left and right sides are determined by the powers of $a^{jg(\xi)}, (j = 0, 1, 2, \ldots)$. After resolving these equations, we get a system of algebraic equation and calculate the values of $k_j(j = 0, 1, 2, \ldots), p, q, r, \omega$ and other variables. Finally, the real constants $k_j(j = 0, 1, 2, \ldots), m, n, l$ and $g(\xi)$ putting into Eq (2.4), yield many TWSs from the Eq (2.1).

Case-1: when $n^2-4ml<0$ and $l \neq 0$, $a^{g(\xi)} = \frac{-n}{2l} + \frac{\sqrt{4ml-n^2}}{2l} \tan\left(\frac{\sqrt{4ml-n^2}}{2}\xi\right)$,

or $a^{g(\xi)} = \frac{-n}{2l} + \frac{\sqrt{4ml-n^2}}{2l} \cot\left(\frac{\sqrt{4ml-n^2}}{2}\xi\right)$.

Case-2: when $n^2-4ml>0$ and $l \neq 0$, $a^{g(\xi)} = \frac{-n}{2l} - \frac{\sqrt{4ml-n^2}}{2l} \tanh\left(\frac{\sqrt{4ml-n^2}}{2}\xi\right)$,

or $a^{g(\xi)} = \frac{-n}{2l} - \frac{\sqrt{4ml-n^2}}{2l} \coth\left(\frac{\sqrt{4ml-n^2}}{2}\xi\right)$.

Case-3: when $n^2 + 4m^2 < 0$, $l \neq 0$ and $l = -m$,

$a^{g(\xi)} = \frac{n}{2m} - \frac{\sqrt{-n^2-4m^2}}{2m} \tan\left(\frac{\sqrt{-n^2-4m^2}}{2}\xi\right)$,

or $a^{g(\xi)} = \frac{n}{2m} + \frac{\sqrt{-n^2-4m^2}}{2m} \cot\left(\frac{\sqrt{-n^2-4m^2}}{2}\xi\right)$.

Case-4: when $n^2 + 4m^2 > 0$, $l \neq 0$ and $l = -m$,

$$a^{g(\xi)} = \frac{n}{2m} + \frac{\sqrt{n^2+4m^2}}{2m}\tanh\left(\frac{\sqrt{n^2+4m^2}}{2}\xi\right),$$

$$\text{or } a^{g(\xi)} = \frac{n}{2m} + \frac{\sqrt{n^2+4m^2}}{2m}\coth\left(\frac{\sqrt{n^2+4m^2}}{2}\xi\right).$$

Case-5: when $n^2-4m^2<0$ and $l = m$, $a^{g(\xi)} = \frac{-n}{2m} - \frac{\sqrt{-n^2+4m^2}}{2m}\tan\left(\frac{\sqrt{-n^2+4m^2}}{2}\xi\right),$

$$\text{or } a^{g(\xi)} = \frac{-n}{2m} - \frac{\sqrt{-n^2+4m^2}}{2m}\cot\left(\frac{\sqrt{-n^2+4m^2}}{2}\xi\right),$$

Case-6: when $n^2+4m^2>0$ and $l = m$, $a^{g(\xi)} = \frac{-n}{2m} - \frac{\sqrt{n^2-4m^2}}{2m}\tanh\left(\frac{\sqrt{n^2-4m^2}}{2}\xi\right),$

$$\text{or } a^{g(\xi)} = \frac{-n}{2m} - \frac{\sqrt{n^2-4m^2}}{2m}\coth\left(\frac{\sqrt{n^2-4m^2}}{2}\xi\right).$$

Case-7: when $n^2 = 4ml$, $a^{g(\xi)} = -\frac{2+n\xi}{2l\xi}$.

Case-8: when $ml<0$, $n = 0$ and $l \neq 0$, $a^{g(\xi)} = -\sqrt{-\frac{m}{l}}\tanh\left(\sqrt{-ml}\xi\right),$

$$\text{or } a^{g(\xi)} = -\sqrt{-\frac{m}{l}}\coth\left(\sqrt{-ml}\xi\right).$$

Case-9: when $n = 0$ and $m = -l$, $a^{g(\xi)} = \frac{1+e^{(-2l\xi)}}{-1+e^{(-2l\xi)}}$.

Case-10: when $m = l = 0$, $a^{g(\xi)} = \cosh(n\xi) + \sinh(n\xi)$.

Case-11: when $m = n = C$ and $l = 0$, $a^{g(\xi)} = e^{C\xi} - 1$.

Case-12: when $n = l = \phi$ and $m = 0$, $a^{g(\xi)} = \frac{e^{\phi\xi}}{1-e^{\phi\xi}}$.

Case-13: when $n = (m + l)$, $a^{g(\xi)} = \frac{1-me^{(m-l)\xi}}{1-le^{(m-l)\xi}}$.

Cas-14: when $n = -(m + l)$, $a^{g(\xi)} = \frac{m-e^{(m-l)\xi}}{1-e^{(m-l)\xi}}$.

Case-15: when $m = 0$, $a^{g(\xi)} = \frac{ne^{n\xi}}{1-le^{n\xi}}$.

Case-16: when $l = n = m \neq 0$, $a^{g(\xi)} = \frac{1}{2}\left(\sqrt{3}\tan\left(\frac{\sqrt{3}}{2}m\xi\right) - 1\right)$.

Case-17: when $l = n = 0$, $a^{g(\xi)} = m\xi$.

Case-18: when $m = n = 0$, $a^{g(\xi)} = \frac{-1}{l\xi}$.

Case-19: when $l = m$ and $n = 0$, $a^{g(\xi)} = \tan(m\xi)$.

Case-20: when $l = 0$, $a^{g(\xi)} = e^{n\xi} - \frac{\phi}{\chi}$.

## 2.2 The unified scheme

In this sub-section, we describe the total procedure to the unified method [52]. Assume that the TWSs to the Eq (2.3) is represented in the following form:

$$u(\xi) = \sum_{j=0}^{S} k_j\varphi(\xi)^j + \sum_{j=1}^{S} l_j\varphi(\xi)^{-j}, \tag{2.6}$$

where $k_j(j = 1,2,\ldots,S)$ and $l_j(j = 1,2,\ldots,S)$, $k_s$ and $l_s$ cannot both be zero simultaneously due to constants that will be examined further. $\varphi = \varphi(\xi)$ satisfy the Riccati differential equation.

$$\varphi'(\xi) = \varphi^2(\xi) + \lambda. \tag{2.7}$$

we determine the positive integer $S$ applying the balancing between the highest order derivatives and highest order nonlinear terms in Eq (2.3). Substituting Eqs (2.6) and (2.7) into Eq (2.3), producing the same powers of $a^j$, $(j = 0,1,2,\ldots)$, then setting each coefficient of $a^j$ be zero yield a set of algebraic equation in terms of $l_j$, $c_j$ and $\lambda$. Substituting $l_j$, $c_j$ and $\lambda$ into (2.6) with the help of (2.7) we obtained the TWSs of Eq (2.1) for various condition of $\lambda$.

Case-01: when $\lambda < 0$,

$$\varphi(\xi) = \begin{cases} \dfrac{\sqrt{-(\mathcal{B}^2 + \mathcal{R}^2)\lambda} - \mathcal{B}\sqrt{-\lambda}\cosh(2\sqrt{-\lambda}(\xi + h))}{\mathcal{B}\sinh(2\sqrt{-\lambda}(\xi + h)) + \mathcal{R}} \\[2mm] \dfrac{-\sqrt{-(\mathcal{B}^2 + \mathcal{R}^2)\lambda} - \mathcal{B}\sqrt{-\lambda}\cosh(2\sqrt{-\lambda}(\xi + h))}{\mathcal{B}\sinh(2\sqrt{-\lambda}(\xi + h)) + \mathcal{R}} \\[2mm] \sqrt{-\lambda} - \dfrac{2\mathcal{B}\sqrt{-\lambda}}{\mathcal{B} + \cosh(2\sqrt{-\lambda}(\xi + h)) - \sinh(2\sqrt{-\lambda}(\xi + h))} \\[2mm] -\sqrt{-\lambda} + \dfrac{2\mathcal{B}\sqrt{-\lambda}}{\mathcal{B} + \cosh(2\sqrt{-\lambda}(\xi + h)) + \sinh(2\sqrt{-\lambda}(\xi + h))} \end{cases}$$

Case-02: when $\lambda > 0$,

$$\varphi(\xi) = \begin{cases} \dfrac{\sqrt{(\mathcal{B}^2 - \mathcal{R}^2)\lambda} - \mathcal{B}\sqrt{\lambda}\cos(2\sqrt{\lambda}(\xi + h))}{\mathcal{B}\sin(2\sqrt{\lambda}(\xi + h)) + \mathcal{R}} \\[2mm] \dfrac{-\sqrt{(\mathcal{B}^2 - \mathcal{R}^2)\lambda} - \mathcal{B}\sqrt{\lambda}\cos(2\sqrt{\lambda}(\xi + h))}{\mathcal{B}\sin(2\sqrt{\lambda}(\xi + h)) + \mathcal{R}} \\[2mm] i\sqrt{\lambda} - \dfrac{2i\mathcal{B}\sqrt{\lambda}}{\mathcal{B} + \cos(2\sqrt{\lambda}(\xi + h)) - i\sin(2\sqrt{\lambda}(\xi + h))} \\[2mm] -i\sqrt{\lambda} + \dfrac{2i\mathcal{B}\sqrt{\lambda}}{\mathcal{B} + \cos(2\sqrt{\lambda}(\xi + h)) + i\sin(2\sqrt{\lambda}(\xi + h))} \end{cases}$$

Case-03: when $\lambda = 0$, $\varphi(\xi) = -\frac{1}{\xi + h}$.

where $\mathcal{B}$ and $\mathcal{R}$ are two real type arbitrary parameters, and $h$ is also arbitrary constant.

## 3. Mathematical formulation of the model

In this section, we will apply the new auxiliary equation method and the unified method to *mKdV−ZK* model to explore TWSs and mathematical analysis. Let us consider the travelling wave transformation

$$u(x, y, z, t) = u(\xi), \ \ \xi = px + qy + rz - \omega t, \tag{3.1}$$

Utilizing Eq (1.1) with the help of Eq (3.1), then we get the form:

$$-\omega u' + p\beta u^2 u' + p^3 u''' + pq^2 u''' + pr^2 u''' = 0, \tag{3.2}$$

Integrating and simplifying of Eq (3.2), the required form:

$$-\omega u + \frac{1}{3}p\beta u^3 + p^3 u'' + pq^2 u'' + pr^2 u'' = 0. \tag{3.3}$$

By using the balancing procedure in (3.3), we find $S = 1$.

### 3.1 Solution analysis through the new auxiliary equation method

We attain the balance value of $S$ from (3.3) the general solution of (2.4) takes the following form:

$$u(\xi) = k_0 + k_1 a^{g(\xi)}, \tag{3.4}$$

where $k_0$, and $k_1$ are constants and to be evaluated latter. By inserting Eq (3.4) and Eq (2.5) into Eq (3.3), and then setting the coefficients of $a^{g(\xi)}$ to zero, we may build a set of algebraic equations that Maple can solve to reach the following solution sets:

Set-1: $\omega = \frac{(4p^2ml - p^2n^2 + 4q^2ml - q^2n^2 + 4r^2ml - r^2n^2)p}{2}$, $k_0 = \sqrt{-\frac{3p^2 + 3q^2 + 3r^2}{2\beta}}n$, $k_1 = 2l\sqrt{-\frac{3p^2 + 3k^2 + 3r^2}{2\beta}}$.

Set-2: $\omega = \frac{(4p^2ml - p^2n^2 + 4q^2ml - q^2n^2 + 4r^2ml - r^2n^2)p}{2}$, $k_0 = -\sqrt{-\frac{3p^2 + 3q^2 + 3m^2}{2\beta}}n$, $k_1 - 2l\sqrt{-\frac{3p^2 + 3q^2 + 3m^2}{2\beta}}$.

Inserting the **Set-1** values in Eq (3.4) along with Eq (3.3), we can attain the following solutions as the *mKdV–ZK* model.

When $n^2 - 4ml < 0$ and $l \neq 0$, $u_1 = \frac{1}{2}\sqrt{-\frac{6(p^2+q^2+r^2)}{\beta}}\sqrt{4ml - n^2}\tan\left(\frac{\sqrt{4ml-n^2}}{2}\xi\right)$,

or $u_2 = -\frac{1}{2}\sqrt{-\frac{6(p^2+q^2+r^2)}{\beta}}\sqrt{4ml - n^2}\cot\left(\frac{\sqrt{4ml-n^2}}{2}\xi\right)$.

When $n^2 - 4ml > 0$ and $l \neq 0$, $u_3 = -\frac{1}{2}\sqrt{-\frac{6(p^2+q^2+r^2)}{\beta}}\sqrt{-4ml + n^2}\tanh\left(\frac{\sqrt{-4ml+n^2}}{2}\xi\right)$,

or $u_4 = -\frac{1}{2}\sqrt{-\frac{6(p^2+q^2+r^2)}{\beta}}\sqrt{-4ml + n^2}\coth\left(\frac{\sqrt{-4ml+n^2}}{2}\xi\right)$.

When $n^2 + 4m^2 < 0$, $l \neq 0$ and $l = -m$,

$u_5 = \frac{1}{2}\sqrt{-\frac{6(p^2+q^2+r^2)}{\beta}}\sqrt{-4m^2 - n^2}\tan\left(\frac{\sqrt{-4m^2-n^2}}{2}\xi\right)$,

or $u_6 = -\sqrt{-\frac{6(p^2+q^2+r^2)}{\beta}}\sqrt{-4m^2 - n^2}\cot\left(\frac{\sqrt{-4m^2-n^2}}{2}\xi\right)$.

When $n^2 + 4m^2 > 0$, $l \neq 0$ and $l = -m$,

$u_7 = -\frac{1}{2}\sqrt{-\frac{6(p^2+q^2+r^2)}{\beta}}\sqrt{4m^2 + n^2}\tanh\left(\frac{\sqrt{4m^2+n^2}}{2}\xi\right)$,

or $u_8 = -\frac{1}{2}\sqrt{-\frac{6(p^2+q^2+r^2)}{\beta}}\sqrt{4m^2 + n^2}\coth\left(\frac{\sqrt{4m^2+n^2}}{2}\xi\right)$.

When $n^2 - 4m^2 < 0$ and $l = m$, $u_9 = \frac{1}{2}\sqrt{-\frac{6(p^2+q^2+r^2)}{\beta}}\sqrt{4m^2 - n^2}\tan\left(\frac{\sqrt{4m^2-n^2}}{2}\xi\right)$,

or $u_{10} = -\frac{1}{2}\sqrt{-\frac{6(p^2+q^2+r^2)}{\beta}}\sqrt{4m^2 - n^2}\cot\left(\frac{\sqrt{4m^2-n^2}}{2}\xi\right)$.

When $n^2 + 4m^2 > 0$ and $l = m$, $u_{11} = -\frac{1}{2}\sqrt{-\frac{6(p^2+q^2+r^2)}{\beta}}\sqrt{-4m^2 + n^2}\tanh\left(\frac{\sqrt{-4m^2+n^2}}{2}\xi\right)$,

or $u_{12} = -\frac{1}{2}\sqrt{-\frac{6(p^2+q^2+r^2)}{\beta}}\sqrt{-4m^2 + n^2}\coth\left(\frac{\sqrt{-4m^2+n^2}}{2}\xi\right)$.

When $n^2 = 4ml$, $u_{13} = -\frac{1}{\xi}\sqrt{-\frac{6(p^2+q^2+r^2)}{\beta}}$.

When $ml < 0$, $n = 0$ and $l \neq 0$, $u_{14} = -l\sqrt{-\frac{6(p^2+q^2+r^2)}{\beta}}\sqrt{-\frac{m}{l}}\tanh\left(\sqrt{-ml}\xi\right)$,

or $u_{15} = -l\sqrt{-\frac{6(p^2+q^2+r^2)}{\beta}}\sqrt{-\frac{m}{l}}\coth\left(\sqrt{-ml}\xi\right)$.

When $n = 0$ and $m = -l$, $u_{16} = l\sqrt{-\frac{6(p^2+q^2+r^2)}{\beta}}\left(\frac{1+e^{(-2l\xi)}}{-1+e^{(-2l\xi)}}\right)$.

When $n = l = \phi$ and $m = 0$, $u_{17} = -\frac{1}{2}\phi\sqrt{-\frac{6(p^2+q^2+r^2)}{\beta}}\left(\frac{1+e^{\phi\xi}}{-1+e^{\phi\xi}}\right)$.

When $n = (m+l)$, $u_{18} = \frac{1}{2}\sqrt{-\frac{6(p^2+q^2+r^2)}{\beta}}(l-m)\left(\frac{le^{(m-l)\xi}+1}{le^{(m-l)\xi}-1}\right)$.

When $n = -(m+l)$, $u_{19} = \frac{1}{2}\sqrt{-\frac{6(p^2+q^2+r^2)}{\beta}}(-m+l)\left(\frac{l+e^{(m-l)\xi}}{-l+e^{(m-l)\xi}}\right)$.

When $m = 0$, $u_{20} = -\frac{1}{2}n\sqrt{-\frac{6(p^2+q^2+r^2)}{\beta}}\left(\frac{1+le^{n\xi}}{-1+le^{n\xi}}\right)$.

When $l = n = m\neq0$, $u_{21} = \frac{1}{2}m\sqrt{-\frac{6(p^2+q^2+r^2)}{\beta}}\left\{\left(\sqrt{3}\tan\left(\frac{\sqrt{3}}{2}m\xi\right)-1\right)+1\right\}$.

When $m = n = 0$, $u_{22} = -\frac{1}{\xi}\sqrt{-\frac{6(p^2+q^2+r^2)}{\beta}}$.

When $l = m$ and $n = 0$, $u_{23} = m\sqrt{-\frac{6(p^2+q^2+r^2)}{\beta}}\tan(m\xi)$.

Inserting the **Set-2** values in Eq (3.4) along with Eq (3.3), we can attain the following solutions as the *mKdV–ZK* model.

When $n^2-4ml<0$ and $l\neq0$, $u_{24} = -\frac{1}{2}\sqrt{-\frac{6(p^2+q^2+r^2)}{\beta}}\sqrt{4ml-n^2}\tan\left(\frac{\sqrt{4ml-n^2}}{2}\xi\right)$,

or $u_{25} = \frac{1}{2}\sqrt{-\frac{6(p^2+q^2+r^2)}{\beta}}\sqrt{4ml-n^2}\cot\left(\frac{\sqrt{4ml-n^2}}{2}\xi\right)$.

When $n^2-4ml>0$ and $l\neq0$, $u_{26} = \frac{1}{2}\sqrt{-\frac{6(p^2+q^2+r^2)}{\beta}}\sqrt{-4ml+n^2}\tanh\left(\frac{\sqrt{-4ml+n^2}}{2}\xi\right)$,

or $u_{27} = \frac{1}{2}\sqrt{-\frac{6(p^2+q^2+r^2)}{\beta}}\sqrt{-4ml+n^2}\coth\left(\frac{\sqrt{-4ml+n^2}}{2}\xi\right)$.

When $n^2+4ml^2<0$, $l\neq0$ and $l = -m$,

$u_{28} = -\frac{1}{2}\sqrt{-\frac{6(p^2+q^2+r^2)}{\beta}}\sqrt{-4m^2-n^2}\tan\left(\frac{\sqrt{-4m^2-n^2}}{2}\xi\right)$,

or $u_{29} = \frac{1}{2}\sqrt{-\frac{6(p^2+q^2+r^2)}{\beta}}\sqrt{-4m^2-n^2}\cot\left(\frac{\sqrt{-4m^2-n^2}}{2}\xi\right)$.

When $n^2+4m^2>0$, $l\neq0$ and $l = -m$,

$u_{30} = \frac{1}{2}\sqrt{-\frac{6(p^2+q^2+r^2)}{\beta}}\sqrt{4m^2+n^2}\tanh\left(\frac{\sqrt{4m^2+n^2}}{2}\xi\right)$,

or $u_{31} = \frac{1}{2}\sqrt{-\frac{6(p^2+q^2+r^2)}{\beta}}\sqrt{4m^2+n^2}\coth\left(\frac{\sqrt{4m^2+n^2}}{2}\xi\right)$.

When $n^2-4m^2<0$ and $l = m$, $u_{32} = -\frac{1}{2}\sqrt{-\frac{6(p^2+q^2+r^2)}{\beta}}\sqrt{4m^2-n^2}\tan\left(\frac{\sqrt{4m^2-n^2}}{2}\xi\right)$,

or $u_{33} = \frac{1}{2}\sqrt{-\frac{6(p^2+q^2+r^2)}{\beta}}\sqrt{4m^2-n^2}\cot\left(\frac{\sqrt{4m^2-n^2}}{2}\xi\right)$.

When $n^2+4m^2>0$ and $l = m$, $u_{34} = \frac{1}{2}\sqrt{-\frac{6(p^2+q^2+r^2)}{\beta}}\sqrt{-4m^2+n^2}\tanh\left(\frac{\sqrt{-4m^2+n^2}}{2}\xi\right)$,

or $u_{35} = \frac{1}{2}\sqrt{-\frac{6(p^2+q^2+r^2)}{\beta}}\sqrt{-4m^2+n^2}\coth\left(\frac{\sqrt{-4m^2+n^2}}{2}\xi\right)$.

When $n^2 = 4ml$, $u_{36} = \frac{1}{\xi}\sqrt{-\frac{6(p^2+q^2+r^2)}{\beta}}$.

When $ml<0$, $n = 0$ and $l\neq0$, $u_{37} = l\sqrt{-\frac{6(p^2+q^2+r^2)}{\beta}}\sqrt{-\frac{m}{l}}\tanh\left(\sqrt{-ml}\xi\right)$,

or $u_{38} = l\sqrt{-\frac{6(p^2+q^2+r^2)}{\beta}}\sqrt{-\frac{m}{l}}\coth\left(\sqrt{-ml}\xi\right)$.

When $n = 0$ and $m = -l$, $u_{39} = -l\sqrt{-\frac{6(p^2+q^2+r^2)}{\beta}}\left(\frac{1+e^{(-2l\xi)}}{-1+e^{(-2l\xi)}}\right)$.

When $n = l = \phi$ and $m = 0$, $u_{40} = \frac{1}{2}\phi\sqrt{-\frac{6(p^2+q^2+r^2)}{\beta}}\left(\frac{1+e^{\phi\xi}}{-1+e^{\phi\xi}}\right)$.

When $n = (m+1)$, $u_{41} = -\frac{1}{2}\sqrt{-\frac{6(p^2+q^2+r^2)}{\beta}}(l-m)\left(\frac{le^{(m-l)\xi}+1}{le^{(m-l)\xi}-1}\right)$.

When $n = -(m+1)$, $u_{42} = \frac{1}{2}\sqrt{-\frac{6(p^2+q^2+r^2)}{\beta}}(-m+l)\left(\frac{l+e^{(m-l)\xi}}{l-e^{(m-l)\xi}}\right)$.

When $m = 0$, $u_{43} = \frac{1}{2}n\sqrt{-\frac{6(p^2+q^2+r^2)}{\beta}}\left(\frac{le^{n\xi}+1}{le^{n\xi}-1}\right)$.

When $l = n = m \neq 0$, $u_{44} = \frac{1}{2}\sqrt{-\frac{6(p^2+q^2+r^2)}{\beta}}\{\left(\sqrt{3}\tan\left(\frac{\sqrt{3}}{2}m\xi\right)-1\right)l+n\}$.

When $m = n = 0$, $u_{45} = \frac{1}{\xi}\sqrt{-\frac{6(p^2+q^2+r^2)}{\beta}}$.

When $l = m$ and $n = 0$, $u_{46} = -m\sqrt{-\frac{6(p^2+q^2+r^2)}{\beta}}\tan(m\xi)$.

## 3.2 Solution analysis through the unified method

Based upon the number of balance principle $S$ the trial solution of Eq (3.3) become of the following form:

$$u(\xi) = k_0 + k_1\varphi(\xi) + l_1\varphi(\xi)^{-1}, \tag{3.5}$$

Where $k_0$, $k_1$ and $l_1$ are arbitrary constant and $k_1$, $l_1$ cannot both be zero simultaneously. Inserting Eq (3.5) and Eq (2.7) into Eq (3.3), and after that, adjusting the $\varphi(\xi)$ factors to zero, we are able to develop following set of collection:

Set-1: $\omega = 2p^3\lambda + 2pq^2\lambda + 2pr^2\lambda$, $k_0 = 0$, $k_1 = \pm M$, $l_1 = 0$.

Set-2: $\omega = -p(6p^2+6q^2+6r^2)\lambda + 2p^3\lambda + 2pq^2\lambda + 2pr^2\lambda, k_0 = 0, k_1 = M, l_1 = M\lambda$.

Set-3: $\omega = p(6p^2+6q^2+6r^2)\lambda + 2p^3\lambda + 2pq^2\lambda + 2pr^2\lambda, k_0 = 0, k_1 = M, k_1 = -M\lambda$.

Set-4: $\omega = p(6p^2+6q^2+6r^2)\lambda + 2p^3\lambda + 2pq^2\lambda + 2pr^2\lambda, k_0 = 0, k_1 = -M, l_1 = M\lambda$.

Set-5: $\omega = -p(6p^2+6q^2+6r^2)\lambda + 2p^3\lambda + 2pq^2\lambda + 2pr^2\lambda, k_0 = 0, k_1 = -M, l_1 = -M\lambda$.

Set-6: $\omega = 2p^3\lambda + 2pq^2\lambda + 2pr^2\lambda, k_0 = 0, k_1 = 0, l_1 = \pm M\lambda$.

Where $M = \sqrt{-\frac{6p^2+6q^2+6r^2}{\beta}}$.

Inserting the **Set-1** values in Eq (3.5) along with Eq (3.3), we can attain the following solutions as the *mKdV−ZK* model.

When $\lambda < 0$, we obtain

$$u_{47} = \frac{\pm M(\sqrt{-(\mathcal{B}^2+\mathcal{R}^2)\lambda} - \mathcal{B}\sqrt{-\lambda}\cosh(2\sqrt{-\lambda}(\xi+h)))}{\mathcal{B}\sinh(2\sqrt{-\lambda}(\xi+h)) + \mathcal{R}}.$$

$$u_{48} = \frac{\pm M(-\sqrt{-(\mathcal{B}^2+\mathcal{R}^2)\lambda} - \mathcal{B}\sqrt{-\lambda}\cosh(2\sqrt{-\lambda}(\xi+h)))}{\mathcal{B}\sinh(2\sqrt{-\lambda}(\xi+h)) + \mathcal{R}}.$$

$$u_{49} = \pm M\left(\sqrt{-\lambda} - \frac{2\mathcal{B}\sqrt{-\lambda}}{\mathcal{B} + \cosh(2\sqrt{-\lambda}(\xi+h)) - \sinh(2\sqrt{-\lambda}(\xi+h))}\right).$$

$$u_{50} = \pm M\left(-\sqrt{-\lambda} + \frac{2\mathcal{B}\sqrt{-\lambda}}{\mathcal{B} + \cosh(2\sqrt{-\lambda}(\xi+h)) + \sinh(2\sqrt{-\lambda}(\xi+h))}\right).$$

When $\lambda > 0$, we obtain

$$u_{51} = \frac{\pm M\left(\sqrt{(\mathcal{B}^2 - \mathcal{R}^2)\lambda} - \mathcal{B}\sqrt{\lambda}\cos(2\sqrt{\lambda}(\xi + h))\right)}{\mathcal{B}\sin(2\sqrt{\lambda}(\xi + h)) + \mathcal{R}}.$$

$$u_{52} = \frac{\pm M\left(-\sqrt{(\mathcal{B}^2 - \mathcal{R}^2)\lambda} - \mathcal{B}\sqrt{\lambda}\cos(2\sqrt{\lambda}(\xi + h))\right)}{\mathcal{B}\sin(2\sqrt{\lambda}(\xi + h)) + \mathcal{R}}.$$

$$u_{53} = \pm M\left(i\sqrt{\lambda} - \frac{2i\mathcal{B}\sqrt{\lambda}}{\mathcal{B} + \cos(2\sqrt{\lambda}(\xi + h)) - i\sin(2\sqrt{\lambda}(\xi + h))}\right).$$

$$u_{54} = \pm M\left(-i\sqrt{\lambda} + \frac{2i\mathcal{B}\sqrt{\lambda}}{\mathcal{B} + \cos(2\sqrt{\lambda}(\xi + h)) + i\sin(2\sqrt{\lambda}(\xi + h))}\right).$$

When $\lambda = 0$, we obtain

$$u_{55} = \frac{\pm M}{(\xi + h)}.$$

Inserting the **Set-2** values in Eq (3.5) along with Eq (3.3), we can attain the following solutions as the *mKdV−ZK* model.

When $\lambda < 0$, we obtain

$$u_{56} = \frac{M\left(\sqrt{-(\mathcal{B}^2 + \mathcal{R}^2)\lambda} - \mathcal{B}\sqrt{-\lambda}\cosh(2\sqrt{-\lambda}(\xi + h))\right)}{\mathcal{B}\sinh(2\sqrt{-\lambda}(\xi + h)) + \mathcal{R}}$$
$$+ \frac{M\lambda(\mathcal{B}\sinh(2\sqrt{-\lambda}(\xi + h)) + \mathcal{R})}{\sqrt{-(\mathcal{B}^2 + \mathcal{R}^2)\lambda} - \mathcal{B}\sqrt{-\lambda}\cosh(2\sqrt{-\lambda}(\xi + h))}.$$

$$u_{57} = \frac{M\left(-\sqrt{-(\mathcal{B}^2 + \mathcal{R}^2)\lambda} - \mathcal{B}\sqrt{-\lambda}\cosh(2\sqrt{-\lambda}(\xi + h))\right)}{\mathcal{B}\sinh(2\sqrt{-\lambda}(\xi + h)) + \mathcal{R}}$$
$$+ \frac{M\lambda(\mathcal{B}\sinh(2\sqrt{-\lambda}(\xi + h)) + \mathcal{R})}{-\sqrt{-(\mathcal{B}^2 + \mathcal{R}^2)\lambda} - \mathcal{B}\sqrt{-\lambda}\cosh(2\sqrt{-\lambda}(\xi + h))}.$$

$$u_{58} = M\sqrt{-\lambda} - \frac{2M\mathcal{B}\sqrt{-\lambda}}{\mathcal{B} + \cosh(2\sqrt{-\lambda}(\xi + h)) - \sinh(2\sqrt{-\lambda}(\xi + h))}$$

$$+ \frac{M\lambda}{\sqrt{-\lambda} - \frac{2\mathcal{B}\sqrt{-\lambda}}{\mathcal{B} + \cosh(2\sqrt{-\lambda}(\xi + h)) - \sinh(2\sqrt{-\lambda}(\xi + h))}}.$$

$$u_{59} = -M\sqrt{-\lambda} + \frac{2M\mathcal{B}\sqrt{-\lambda}}{\mathcal{B} + \cosh(2\sqrt{-\lambda}(\xi + h)) + \sinh(2\sqrt{-\lambda}(\xi + h))}$$

$$+ \frac{M\lambda}{-\sqrt{-\lambda} + \frac{2\mathcal{B}\sqrt{-\lambda}}{\mathcal{B} + \cosh(2\sqrt{-\lambda}(\xi + h)) + \sinh(2\sqrt{-\lambda}(\xi + h))}}.$$

When $\lambda > 0$, we obtain

$$u_{60} = \frac{M(\sqrt{((\mathcal{B}^2 - \mathcal{R}^2)\lambda} - \mathcal{B}\sqrt{\lambda}\cos(2\sqrt{\lambda}(\xi + h)))}{\mathcal{B}\sin(2\sqrt{\lambda}(\xi + h)) + \mathcal{R}}$$

$$+ \frac{M\lambda(\mathcal{B}\sin(2\sqrt{\lambda}(\xi + h)) + \mathcal{R})}{\sqrt{((\mathcal{B}^2 - \mathcal{R}^2)\lambda} - \mathcal{B}\sqrt{\lambda}\cos(2\sqrt{\lambda}(\xi + h))}.$$

$$u_{61} = \frac{M(-\sqrt{((\mathcal{B}^2 - \mathcal{R}^2)\lambda} - \mathcal{B}\sqrt{\lambda}\cos(2\sqrt{\lambda}(\xi + h)))}{\mathcal{B}\sin(2\sqrt{\lambda}(\xi + h)) + \mathcal{R}}$$

$$+ \frac{M\lambda(\mathcal{B}\sin(2\sqrt{\lambda}(\xi + h)) + \mathcal{R})}{-\sqrt{((\mathcal{B}^2 - \mathcal{R}^2)\lambda} - \mathcal{B}\sqrt{\lambda}\cos(2\sqrt{\lambda}(\xi + h))}.$$

$$u_{62} = M\left( i\sqrt{\lambda} - \frac{2i\mathcal{B}\sqrt{\lambda}}{\mathcal{B} + \cos(2\sqrt{\lambda}(\xi + h)) - i\sin(2\sqrt{\lambda}(\xi + h))} \right)$$

$$+ \frac{M\lambda}{\left( i\sqrt{\lambda} - \frac{2i\mathcal{B}\sqrt{\lambda}}{\mathcal{B} + \cos(2\sqrt{\lambda}(\xi + h)) - i\sin(2\sqrt{\lambda}(\xi + h))} \right)}.$$

$$u_{63} = M\left( -i\sqrt{\lambda} + \frac{2i\mathcal{B}\sqrt{\lambda}}{\mathcal{B} + \cos(2\sqrt{\lambda}(\xi + h)) + i\sin(2\sqrt{\lambda}(\xi + h))} \right)$$

$$+ \frac{M\lambda}{\left( -i\sqrt{\lambda} + \frac{2i\mathcal{B}\sqrt{\lambda}}{\mathcal{B} + \cos(2\sqrt{\lambda}(\xi + h)) + i\sin(2\sqrt{\lambda}(\xi + h))} \right)}.$$

When $\lambda = 0$, we obtain

$$u_{64} = -\frac{M}{\xi + h} - M\lambda(\xi + h).$$

Inserting the **Set-3** values in Eq (3.5) along with Eq (3.3), we can attain the following solutions as the *mKdV−ZK* model.

When $\lambda < 0$, we obtain

$$u_{65} = \frac{M(\sqrt{-(\mathcal{B}^2 + \mathcal{R}^2)\lambda} - \mathcal{B}\sqrt{-\lambda}\cosh(2\sqrt{-\lambda}(\xi + h)))}{\mathcal{B}\sinh(2\sqrt{-\lambda}(\xi + h)) + \mathcal{R}}$$
$$- \frac{M\lambda(\mathcal{B}\sinh(2\sqrt{-\lambda}(\xi + h)) + \mathcal{R})}{\sqrt{-(\mathcal{B}^2 + \mathcal{R}^2)\lambda} - \mathcal{B}\sqrt{-\lambda}\cosh(2\sqrt{-\lambda}(\xi + h))}.$$

$$u_{66} = \frac{M(-\sqrt{-(\mathcal{B}^2 + \mathcal{R}^2)\lambda} - \mathcal{B}\sqrt{-\lambda}\cosh(2\sqrt{-\lambda}(\xi + h)))}{\mathcal{B}\sinh(2\sqrt{-\lambda}(\xi + h)) + \mathcal{R}}$$
$$- \frac{M\lambda(\mathcal{B}\sinh(2\sqrt{-\lambda}(\xi + h)) + \mathcal{R})}{-\sqrt{-(\mathcal{B}^2 + \mathcal{R}^2)\lambda} - \mathcal{B}\sqrt{-\lambda}\cosh(2\sqrt{-\lambda}(\xi + h))}.$$

$$u_{67} = M\sqrt{-\lambda} - \frac{2M\mathcal{B}\sqrt{-\lambda}}{\mathcal{B} + \cosh(2\sqrt{-\lambda}(\xi + h)) - \sinh(2\sqrt{-\lambda}(\xi + h))}$$
$$- \frac{M\lambda}{\sqrt{-\lambda} - \frac{2\mathcal{B}\sqrt{-\lambda}}{\mathcal{B} + \cosh(2\sqrt{-\lambda}(\xi + h)) - \sinh(2\sqrt{-\lambda}(\xi + h))}}.$$

$$u_{68} = -M\sqrt{-\lambda} + \frac{2M\mathcal{B}\sqrt{-\lambda}}{\mathcal{B} + \cosh(2\sqrt{-\lambda}(\xi + h)) + \sinh(2\sqrt{-\lambda}(\xi + h))}$$
$$- \frac{M\lambda}{-\sqrt{-\lambda} + \frac{2\mathcal{B}\sqrt{-\lambda}}{\mathcal{B} + \cosh(2\sqrt{-\lambda}(\xi + h)) + \sinh(2\sqrt{-\lambda}(\xi + h))}}.$$

When $\lambda > 0$, we obtain

$$u_{69} = \frac{M(\sqrt{((\mathcal{B}^2 - \mathcal{R}^2)\lambda} - \mathcal{B}\sqrt{\lambda}\cos(2\sqrt{\lambda}(\xi + h)))}{\mathcal{B}\sin(2\sqrt{\lambda}(\xi + h)) + \mathcal{R}}$$
$$- \frac{M\lambda(\mathcal{B}\sin(2\sqrt{\lambda}(\xi + h)) + \mathcal{R})}{\sqrt{((\mathcal{B}^2 - \mathcal{R}^2)\lambda} - \mathcal{B}\sqrt{\lambda}\cos(2\sqrt{\lambda}(\xi + h))}.$$

$$u_{70} = \frac{M(-\sqrt{((\mathcal{B}^2 - \mathcal{R}^2)\lambda} - \mathcal{B}\sqrt{\lambda}\cos(2\sqrt{\lambda}(\xi + h)))}{\mathcal{B}\sin(2\sqrt{\lambda}(\xi + h)) + \mathcal{R}}$$
$$- \frac{M\lambda(\mathcal{B}\sin(2\sqrt{\lambda}(\xi + h)) + \mathcal{R})}{-\sqrt{((\mathcal{B}^2 - \mathcal{R}^2)\lambda} - \mathcal{B}\sqrt{\lambda}\cos(2\sqrt{\lambda}(\xi + h))}.$$

$$u_{71} = M\left(i\sqrt{\lambda} - \frac{2i\mathcal{B}\sqrt{\lambda}}{\mathcal{B} + \cos(2\sqrt{\lambda}(\xi + h)) - i\sin(2\sqrt{\lambda}(\xi + h))}\right)$$
$$- \frac{M\lambda}{\left(i\sqrt{\lambda} - \frac{2i\mathcal{B}\sqrt{\lambda}}{\mathcal{B} + \cos(2\sqrt{\lambda}(\xi + h)) - i\sin(2\sqrt{\lambda}(\xi + h))}\right)}.$$

$$u_{72} = M\left(-i\sqrt{\lambda} + \frac{2i\mathcal{B}\sqrt{\lambda}}{\mathcal{B} + \cos(2\sqrt{\lambda}(\xi + h)) + i\sin(2\sqrt{\lambda}(\xi + h))}\right)$$
$$- \frac{M\lambda}{\left(-i\sqrt{\lambda} + \frac{2i\mathcal{B}\sqrt{\lambda}}{\mathcal{B} + \cos(2\sqrt{\lambda}(\xi + h)) + i\sin(2\sqrt{\lambda}(\xi + h))}\right)}.$$

When $\lambda = 0$, we obtain

$$u_{73} = -\frac{M}{\xi + h} + M\lambda(\xi + h).$$

Inserting the **Set-4** values in Eq (3.5) along with Eq (3.3), we can attain the following solutions as the *mKdV−ZK* model.

When $\lambda < 0$, we obtain

$$u_{74} = -\frac{M(\sqrt{-(\mathcal{B}^2 + \mathcal{R}^2)\lambda} - \mathcal{B}\sqrt{-\lambda}\cosh(2\sqrt{-\lambda}(\xi + h)))}{\mathcal{B}\sinh(2\sqrt{-\lambda}(\xi + h)) + \mathcal{R}}$$
$$+ \frac{M\lambda(\mathcal{B}\sinh(2\sqrt{-\lambda}(\xi + h)) + \mathcal{R})}{\sqrt{-(\mathcal{B}^2 + \mathcal{R}^2)\lambda} - \mathcal{B}\sqrt{-\lambda}\cosh(2\sqrt{-\lambda}(\xi + h))}.$$

$$u_{75} = -\frac{M(-\sqrt{-(\mathcal{B}^2 + \mathcal{R}^2)\lambda} - \mathcal{B}\sqrt{-\lambda}\cosh(2\sqrt{-\lambda}(\xi + h)))}{\mathcal{B}\sinh(2\sqrt{-\lambda}(\xi + h)) + \mathcal{R}}$$
$$+ \frac{M\lambda(\mathcal{B}\sinh(2\sqrt{-\lambda}(\xi + h)) + \mathcal{R})}{-\sqrt{-(\mathcal{B}^2 + \mathcal{R}^2)\lambda} - \mathcal{B}\sqrt{-\lambda}\cosh(2\sqrt{-\lambda}(\xi + h))}.$$

$$u_{76} = -M\sqrt{-\lambda} + \frac{2M\mathcal{B}\sqrt{-\lambda}}{\mathcal{B} + \cosh(2\sqrt{-\lambda}(\xi + h)) - \sinh(2\sqrt{-\lambda}(\xi + h))}$$
$$+ \frac{M\lambda}{\sqrt{-\lambda} - \frac{2\mathcal{B}\sqrt{-\lambda}}{\mathcal{B} + \cosh(2\sqrt{-\lambda}(\xi + h)) - \sinh(2\sqrt{-\lambda}(\xi + h))}}.$$

$$u_{77} = M\sqrt{-\lambda} - \frac{2M\mathcal{B}\sqrt{-\lambda}}{\mathcal{B} + \cosh(2\sqrt{-\lambda}(\xi + h)) + \sinh(2\sqrt{-\lambda}(\xi + h))}$$
$$+ \frac{M\lambda}{-\sqrt{-\lambda} + \frac{2\mathcal{B}\sqrt{-\lambda}}{\mathcal{B} + \cosh(2\sqrt{-\lambda}(\xi + h)) + \sinh(2\sqrt{-\lambda}(\xi + h))}}.$$

When $\lambda > 0$, we obtain

$$
\begin{aligned}
u_{78} = & -\frac{M(\sqrt{((\mathcal{B}^2 - \mathcal{R}^2)\lambda} - \mathcal{B}\sqrt{\lambda}\cos(2\sqrt{\lambda}(\xi + h)))}{\mathcal{B}\sin(2\sqrt{\lambda}(\xi + h)) + \mathcal{R}} \\
& + \frac{M\lambda(\mathcal{B}\sin(2\sqrt{\lambda}(\xi + h)) + \mathcal{R})}{\sqrt{((\mathcal{B}^2 - \mathcal{R}^2)\lambda} - \mathcal{B}\sqrt{\lambda}\cos(2\sqrt{\lambda}(\xi + h))} \cdot
\end{aligned}
$$

$$
\begin{aligned}
u_{79} = & -\frac{M(-\sqrt{((\mathcal{B}^2 - \mathcal{R}^2)\lambda} - \mathcal{B}\sqrt{\lambda}\cos(2\sqrt{\lambda}(\xi + h)))}{\mathcal{B}\sin(2\sqrt{\lambda}(\xi + h)) + \mathcal{R}} \\
& + \frac{M\lambda(\mathcal{B}\sin(2\sqrt{\lambda}(\xi + h)) + \mathcal{R})}{-\sqrt{((\mathcal{B}^2 - \mathcal{R}^2)\lambda} - \mathcal{B}\sqrt{\lambda}\cos(2\sqrt{\lambda}(\xi + h))} \cdot
\end{aligned}
$$

$$
\begin{aligned}
u_{80} = & -Mi\sqrt{\lambda} + \frac{2iM\mathcal{B}\sqrt{\lambda}}{\mathcal{B} + \cos(2\sqrt{\lambda}(\xi + h)) - i\sin(2\sqrt{\lambda}(\xi + h))} \\
& + \frac{M\lambda}{\left(i\sqrt{\lambda} - \frac{2i\mathcal{B}\sqrt{\lambda}}{\mathcal{B} + \cos(2\sqrt{\lambda}(\xi + h)) - i\sin(2\sqrt{\lambda}(\xi + h))}\right)} \cdot
\end{aligned}
$$

$$
\begin{aligned}
u_{81} = & Mi\sqrt{\lambda} - \frac{2iM\mathcal{B}\sqrt{\lambda}}{\mathcal{B} + \cos(2\sqrt{\lambda}(\xi + h)) + i\sin(2\sqrt{\lambda}(\xi + h))} \\
& + \frac{M\lambda}{\left(-i\sqrt{\lambda} + \frac{2i\mathcal{B}\sqrt{\lambda}}{\mathcal{B} + \cos(2\sqrt{\lambda}(\xi + h)) + i\sin(2\sqrt{\lambda}(\xi + h))}\right)} \cdot
\end{aligned}
$$

When $\lambda = 0$, we obtain

$$
u_{82} = \frac{M}{\xi + h} - M\lambda(\xi + h).
$$

Inserting the **Set-5** values in Eq (3.5) along with Eq (3.3), we can attain the following solutions as the *mKdV−ZK* model.

When $\lambda < 0$, we obtain

$$
\begin{aligned}
u_{83} = & -\frac{M(\sqrt{-(\mathcal{B}^2 + \mathcal{R}^2)\lambda} - \mathcal{B}\sqrt{-\lambda}\cosh(2\sqrt{-\lambda}(\xi + h)))}{\mathcal{B}\sinh(2\sqrt{-\lambda}(\xi + h)) + \mathcal{R}} \\
& - \frac{M\lambda(\mathcal{B}\sinh(2\sqrt{-\lambda}(\xi + h)) + \mathcal{R})}{\sqrt{-(\mathcal{B}^2 + \mathcal{R}^2)\lambda} - \mathcal{B}\sqrt{-\lambda}\cosh(2\sqrt{-\lambda}(\xi + h))} \cdot
\end{aligned}
$$

$$u_{84} = -\frac{M(-\sqrt{-(\mathcal{B}^2 + \mathcal{R}^2)\lambda} - \mathcal{B}\sqrt{-\lambda}\cosh(2\sqrt{-\lambda}(\xi + h)))}{\mathcal{B}\sinh(2\sqrt{-\lambda}(\xi + h)) + \mathcal{R}}$$
$$-\frac{M\lambda(\mathcal{B}\sinh(2\sqrt{-\lambda}(\xi + h)) + \mathcal{R})}{-\sqrt{-(\mathcal{B}^2 + \mathcal{R}^2)\lambda} - \mathcal{B}\sqrt{-\lambda}\cosh(2\sqrt{-\lambda}(\xi + h))}.$$

$$u_{85} = -M\sqrt{-\lambda} + \frac{2M\mathcal{B}\sqrt{-\lambda}}{\mathcal{B} + \cosh(2\sqrt{-\lambda}(\xi + h)) - \sinh(2\sqrt{-\lambda}(\xi + h))}$$
$$-\frac{M\lambda}{\sqrt{-\lambda} - \frac{2\mathcal{B}\sqrt{-\lambda}}{\mathcal{B} + \cosh(2\sqrt{-\lambda}(\xi + h)) - \sinh(2\sqrt{-\lambda}(\xi + h))}}.$$

$$u_{86} = M\sqrt{-\lambda} - \frac{2M\mathcal{B}\sqrt{-\lambda}}{\mathcal{B} + \cosh(2\sqrt{-\lambda}(\xi + h)) + \sinh(2\sqrt{-\lambda}(\xi + h))}$$
$$-\frac{M\lambda}{-\sqrt{-\lambda} + \frac{2\mathcal{B}\sqrt{-\lambda}}{\mathcal{B} + \cosh(2\sqrt{-\lambda}(\xi + h)) + \sinh(2\sqrt{-\lambda}(\xi + h))}}.$$

When $\lambda > 0$, we obtain

$$u_{87} = -\frac{M(\sqrt{((\mathcal{B}^2 - \mathcal{R}^2)\lambda} - \mathcal{B}\sqrt{\lambda}\cos(2\sqrt{\lambda}(\xi + h)))}{\mathcal{B}\sin(2\sqrt{\lambda}(\xi + h)) + \mathcal{R}}$$
$$-\frac{M\lambda(\mathcal{B}\sin(2\sqrt{\lambda}(\xi + h)) + \mathcal{R})}{\sqrt{((\mathcal{B}^2 - \mathcal{R}^2)\lambda} - \mathcal{B}\sqrt{\lambda}\cos(2\sqrt{\lambda}(\xi + h))}.$$

$$u_{88} = -\frac{M(-\sqrt{((\mathcal{B}^2 - \mathcal{R}^2)\lambda} - \mathcal{B}\sqrt{\lambda}\cos(2\sqrt{\lambda}(\xi + h)))}{\mathcal{B}\sin(2\sqrt{\lambda}(\xi + h)) + \mathcal{R}}$$
$$-\frac{M\lambda(\mathcal{B}\sin(2\sqrt{\lambda}(\xi + h)) + \mathcal{R})}{-\sqrt{((\mathcal{B}^2 - \mathcal{R}^2)\lambda} - \mathcal{B}\sqrt{\lambda}\cos(2\sqrt{\lambda}(\xi + h))}.$$

$$u_{89=} - Mi\sqrt{\lambda} + \frac{2iM\mathcal{B}\sqrt{\lambda}}{\mathcal{B} + \cos(2\sqrt{\lambda}(\xi + h)) - i\sin(2\sqrt{\lambda}(\xi + h))}$$
$$-\frac{M\lambda}{\left(i\sqrt{\lambda} - \frac{2i\mathcal{B}\sqrt{\lambda}}{\mathcal{B} + \cos(2\sqrt{\lambda}(\xi + h)) - i\sin(2\sqrt{\lambda}(\xi + h))}\right)}.$$

$$u_{90} = Mi\sqrt{\lambda} - \frac{2iM\mathcal{B}\sqrt{\lambda}}{\mathcal{B} + \cos(2\sqrt{\lambda}(\xi + h)) + i\sin(2\sqrt{\lambda}(\xi + h))}$$
$$-\frac{M\lambda}{\left(-i\sqrt{\lambda} + \frac{2i\mathcal{B}\sqrt{\lambda}}{\mathcal{B} + \cos(2\sqrt{\lambda}(\xi + h)) + i\sin(2\sqrt{\lambda}(\xi + h))}\right)}.$$

When $\lambda = 0$, we obtain

$$u_{91} = \frac{M}{\xi + h} + M\lambda(\xi + h).$$

Inserting the **Set-6** values in Eq (3.5) along with Eq (3.3), we can attain the following solutions as the *mKdV−ZK* model.

When $\lambda < 0$, we obtain

$$u_{92} = \frac{\pm M\lambda(\mathcal{B}\sin(2\sqrt{-\lambda}(\xi + h)) + \mathcal{R})}{\sqrt{-((\mathcal{B}^2 + \mathcal{R}^2)\lambda - \mathcal{B}\sqrt{-\lambda}\cos(2\sqrt{-\lambda}(\xi + h))}}.$$

$$u_{93} = \frac{\pm M\lambda(\mathcal{B}\sin(2\sqrt{-\lambda}(\xi + h)) + \mathcal{R})}{-\sqrt{-((\mathcal{B}^2 + \mathcal{R}^2)\lambda - \mathcal{B}\sqrt{-\lambda}\cos(2\sqrt{-\lambda}(\xi + h))}}.$$

$$u_{94} = \frac{\pm M\lambda}{\sqrt{-\lambda} - \frac{2\mathcal{B}\sqrt{-\lambda}}{\mathcal{B}+\cosh(2\sqrt{-\lambda}(\xi+h))-\sin(2\sqrt{-\lambda}(\xi+h))}}.$$

$$u_{95} = \frac{\pm M\lambda}{-\sqrt{-\lambda} + \frac{2\mathcal{B}\sqrt{-\lambda}}{\mathcal{B}+\cosh(2\sqrt{-\lambda}(\xi+h))+\sin(2\sqrt{-\lambda}(\xi+h))}}.$$

When $\lambda > 0$, we obtain

$$u_{96} = \frac{\pm M\lambda(\mathcal{B}\sin(2\sqrt{\lambda}(\xi + h)) + \mathcal{R})}{\sqrt{((\mathcal{B}^2 - \mathcal{R}^2)\lambda - \mathcal{B}\sqrt{\lambda}\cos(2\sqrt{\lambda}(\xi + h))}}.$$

$$u_{97} = \frac{\pm M\lambda(\mathcal{B}\sin(2\sqrt{\lambda}(\xi + h)) + \mathcal{R})}{-\sqrt{((\mathcal{B}^2 - \mathcal{R}^2)\lambda - \mathcal{B}\sqrt{\lambda}\cos(2\sqrt{\lambda}(\xi + h))}}.$$

$$u_{98} = \frac{\pm M\lambda}{i\sqrt{\lambda} - \frac{2i\mathcal{B}\sqrt{\lambda}}{\mathcal{B}+\cos(2\sqrt{\lambda}(\xi+h))-i\sin(2\sqrt{\lambda}(\xi+h))}}.$$

$$u_{99} = \frac{\pm M\lambda}{i\sqrt{\lambda} - \frac{2i\mathcal{B}\sqrt{\lambda}}{\mathcal{B}+\cos(2\sqrt{\lambda}(\xi+h))-i\sin(2\sqrt{\lambda}(\xi+h))}}.$$

When $\lambda = 0$, we obtain

$$u_{100} = \mp M\lambda(\xi + h).$$

## 4. Result and discussion

This part is divided into two subparts. Subpart 4.1 shows a graphical illustration of the obtained solutions, and Subpart 4.2 provides a physical description. Using MATLAB software, 3D, Contour, and combined 2D wave profiles are made by choosing an appropriate undefined parameter in the (3+1)-dimensional mKdV-ZK model. Functions with two-dimensional input and one-dimensional output can be displayed using three-dimensional graphs. In data

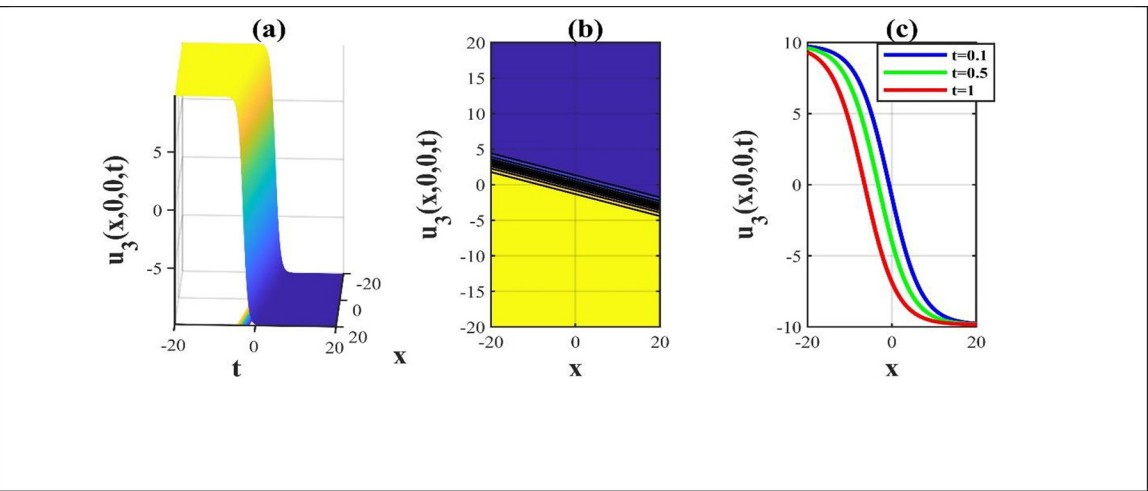

**Fig 1. The depictions of the solution $u_3(x,y,z,t)$ for picking unrestricted parameters and we construct a 3D, contour and amalgamated 2D representation over the rang $-20{\leq}x,t{\leq}20$.**

analysis, these graphs are frequently used to identify the highest and lowest levels in a multidimensional data collection. This part aims to illustrate the solutions discovered during this research. We study the physical use of the wave.

## 4.1 Graphically illustration

See Figs 1–8.

## 4.2. Physical description

Our main focus was on the wave function $u(x,y,z,t)$ solutions to the (3+1)-dimensional *mKdV* $-ZK$ models and how the wave profiles are influenced by the parameters. The steady propagation of each wave is depicted in 3D, contour, and combination 2D images. The 3D

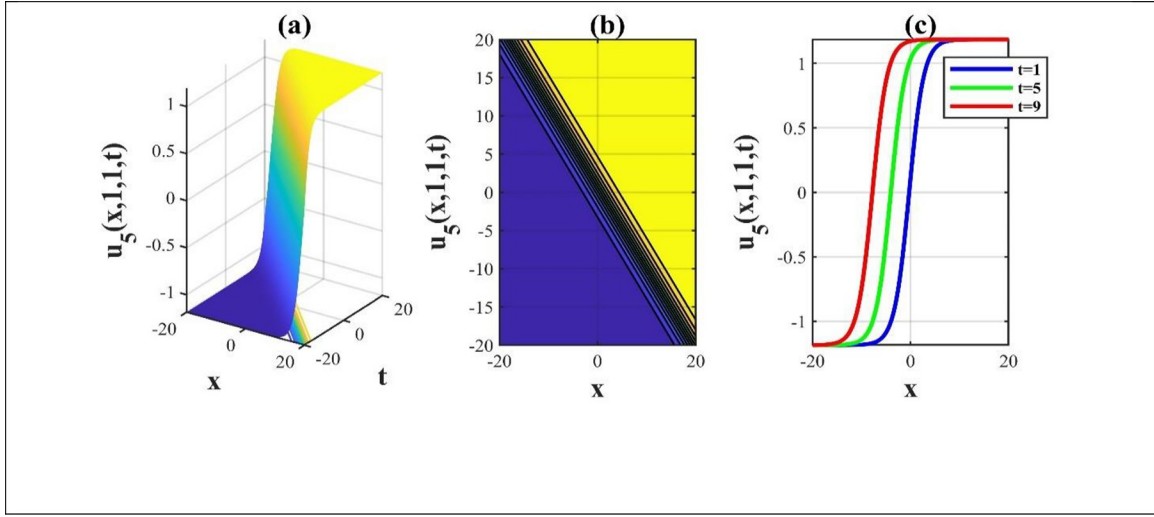

**Fig 2. The illustrations of the solution $u_5(x,y,z,t)$ opting for unrestricted constrains and we get a 3D, contour and amalgamated 2D diagram over the rang $-20{\leq}x,t{\leq}20$.**

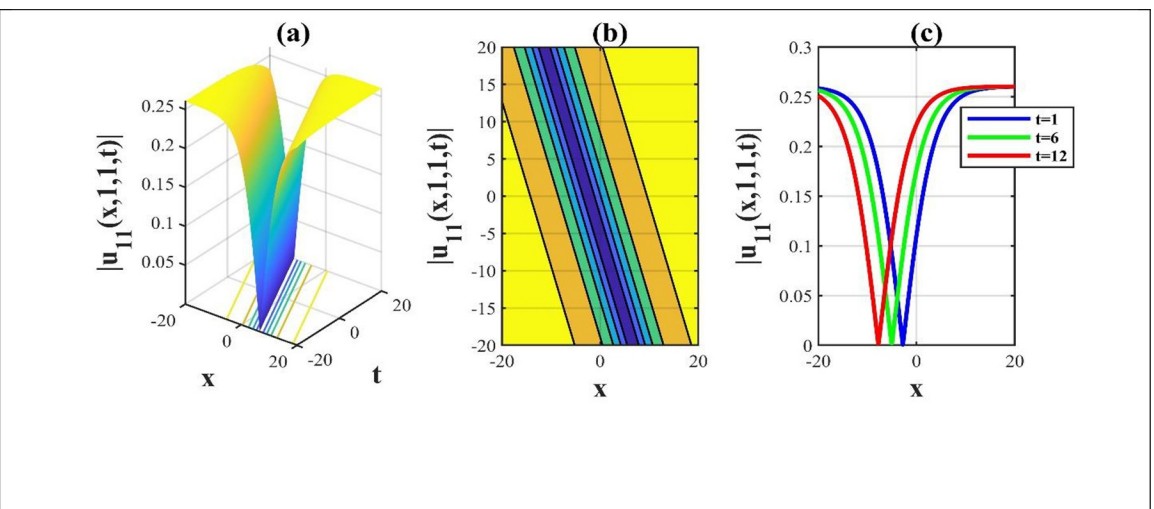

**Fig 3. The modulus portrayed of the solution $u_{11}(x,y,z,t)$ for electing free parameters and we attain a 3D, contour and amalgamated 2D diagram over the rang $-20 \leq x, t \leq 20$.**

representation of the solution $u_3(x,y,z,t)$ for the choosing values of parameters of $p = 0.3$, $q = 1$, $r = 1$, $m = 0.3$, $n = 2$, $l = 0.7$, $\beta = -0.1$, $y = z = 0$. The 3D structure presents the flat kink shaped soliton of this solution, which is portrayed in Fig 1(A) and involved contour in Fig 1(B) is plotted. Fig 1(C) demonstrates the progression of the waves for different values of $t = 0.1, 0.1, 1$. The desired solution $u_5(x,y,z,t)$ highlights flat anti-kink shaped soliton for taking free parameter of $p = -1$, $q = 0.3$, $r = -0.1$, $m = 0.9$, $n = 1$, $l = -0.9$, $\beta = -1$, $y = z = 1$ as seen in Fig 2(A) and equivalent contour in Fig 2(B) are plotted respectively. Fig 2(C) indicates the progression of the waves for various values of $t = 0.3, 0.9, 1.5$. The 3D representation of the modulus solution $u_{11}(x,y,z,t)$ for the suitable values of parameters of $p = 0.3$, $q = 0.2$, $r = 0.5$, $m = 0.25$, $n = 1.2$, $l = 0.25$, $\beta = -10$, $y = z = 1$. The 3D structure presents the dark soliton or V-shaped soliton of this solution, which is displayed in Fig 3(A) and associated contour in Fig 3(B) is plotted. Fig 3

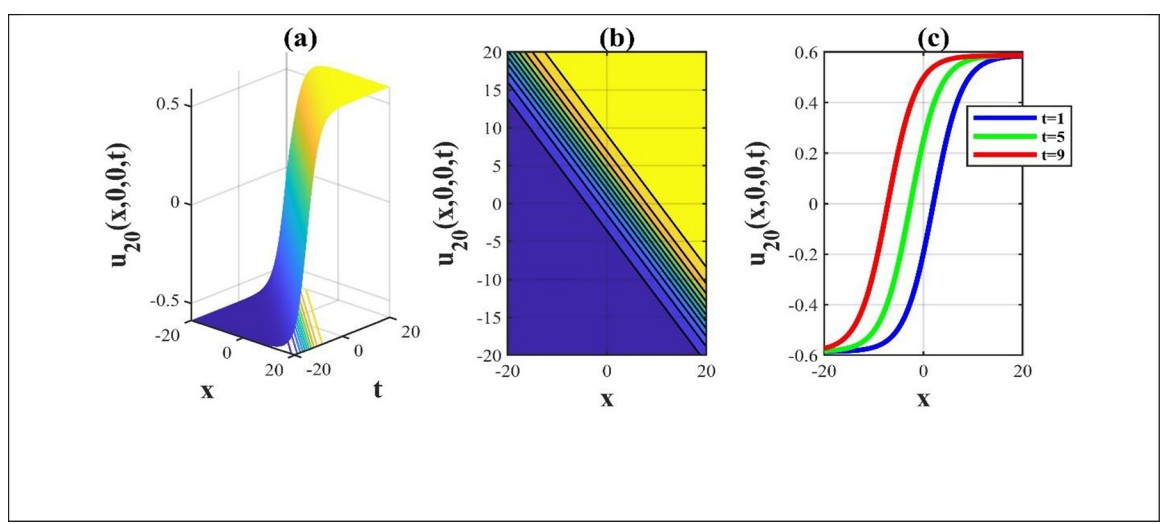

**Fig 4. The attributes of the solution $u_{20}(x,y,z,t)$ for picking unrestricted parameters and we build a 3D, contour and combined 2D a schematic over the rang $-20 \leq x, t \leq 20$.**

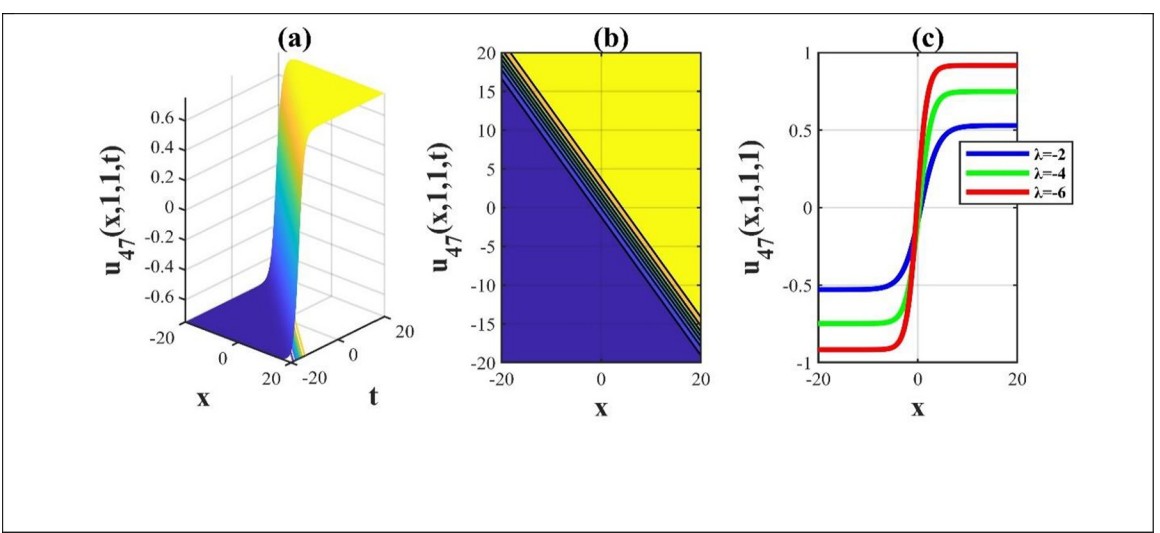

**Fig 5. The diagrams of the solution $u_{47}(x,y,z,t)$ for determining parameters and we create a 3D, contour and amalgamated 2D diagram over the rang $-20{\leq}x,t{\leq}20$.**

(C) shows the progression of the waves for different values of $t = 1, 5, 9$. The 2D plot provides a better understanding the effect of free parameters for different values of $t$ and shows various positions of these values. We depict the 3D wave structure of the solution $u_{20}(x,y,z,t)$ for the parameters of $p = 0.7, q = -0.3, r = 2, m = -2.5, n = -5.5, l = -3, \beta = -5, y = z = 0$. The 3D structure represent the smooth anti-kink shaped soliton of this solution, which is portrayed in Fig 4(A) and associated contour in Fig 4(B) is plotted. Fig 4(C) demonstrates the progression of the waves for several values of $t = 1, 5, 9$. The 3D surface of the solution $u_{55}(x,y,z,t)$ conveys the smooth anti-kink soliton for the selecting parametric values of $k = -0.2, l = 0.3, m = 0.1, h = 0, b = 2, r = 1, h = 0, \beta = -6, \lambda = -4, y = z = 1$. The 3D structure is displayed in Fig 5(A) and its corresponding contour in Fig 5(B) is plotted. Also, Fig 5(C) shows

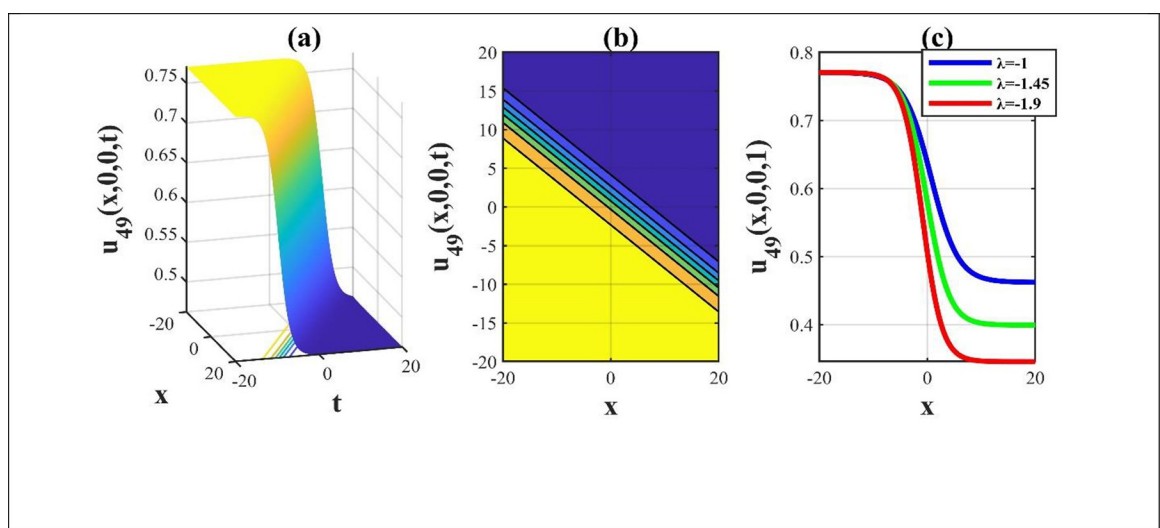

**Fig 6. The sketches of the solution $u_{49}(x,y,z,t)$ for picking unrestricted parameters and we construct a 3D, contour and amalgamated 2D representation over the rang $-20{\leq}x,t{\leq}20$.**

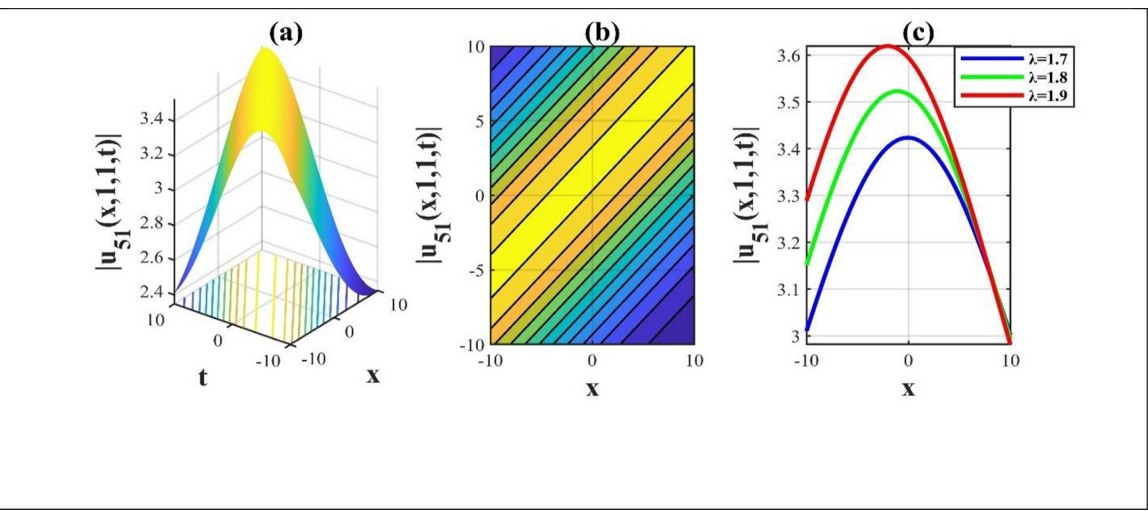

**Fig 7. The modulus attributes of the solution $u_{51}(x,y,z,t)$ for choosing parameters and we construct a 3D, contour and combined 2D representation over the rang $-10 \leq x,t \leq 10$.**

the progression of the waves for distinguished values of $\lambda = -2,-4,-6$. The desired solution $u_{57}(x,y,z,t)$ shows smooth kink shaped soliton for picking unrestricted parameter of $k = 0.2, l = 0.9, m = 0.2, h = -0.5, b = 0.2, \beta = -9, \lambda = -1, y = z = 0$, is depicted in Fig 6 (A). and related contour in Fig 6(B) are plotted respectively. Fig 6(C) demonstrates the progression of the waves for distinct values of $\lambda = -1,-2,-3$. The 3D wave structure of the modulus solution $u_{59}(x,y,z,t)$ signifies the bell shaped soliton (which another name bright soliton studying in optical fiber) for the free parameter of $k = 0.055, \; l = 0.47, m = 0.4, h = 1, b = 1, r = 5, \lambda = 1.8, y = z = 1$. The 3D structure is showed in Fig 7(A) and associated contour in Fig 7 (B) is plotted. Fig 7(C) shows the progression of the waves for different values of $t = 1.7, 1.8, 1.9$. The absolute solution $u_{62}(x,y,z,t)$ displays M-shaped soliton for the free parameter of $k =$

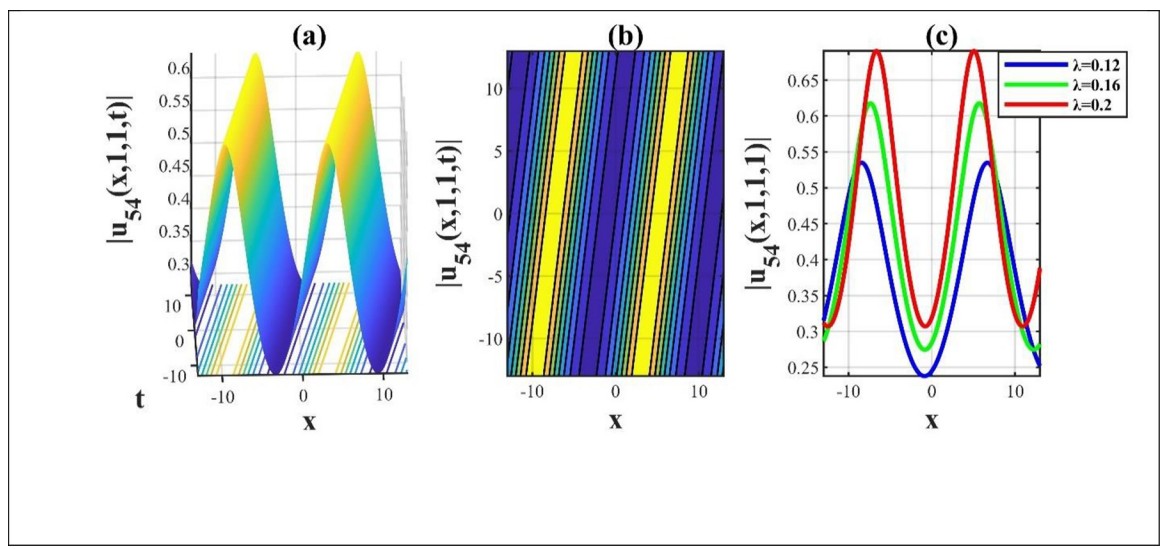

**Fig 8. The modulus illustrations of the solution $u_{54}(x,y,z,t)$ for taking free parameters and we attain a 3D, contour and combined 2D diagram over the rang $-10 \leq x,t \leq 10$.**

$0.6, l = 0.1, m = 0.4, h = 0, b = 5, r = 2, \beta = -3, \lambda = 0.17, y = z = 1$ is displayed in Fig 8 (A). and associated contour in Fig 8(B) are plotted respectively. Fig 8(C) shows the progression of the waves for different values of $\lambda = 0.12, 0.16, 0.2$. A stable and durable wave solution is the M-shaped soliton. It can be provide far-reaching data transmission. The impact of nonlinear parameters $\beta$ and $\lambda$ of the obtained solution based on the time exhibits the wave amplitude of the horizontal axis.

### 4.3 Comparison

In this research article, we will discuss comparison between attained solutions and Al-Ghafri et. al [51] solutions. Al-Ghafri et. al [51] studied of the (3+1)-dimensional space–time fractional *mKdV−ZK* equation by the variable separated ODE method. If we consider $\alpha = 1$, the (3+1)-dimensional space-time fractional *mKdV−ZK* model will be converted (3+1)-dimensional *mKdV−ZK* model. Using the variable separated ODE method, Al-Ghafri et. al [51] have explored Two case included twelve subcase that contains fifty-four solitons' solutions. On the other hand, the new auxiliary equation method used to generate many wave solutions for the (3+1)-dimensional *mKdV−ZK* Model. Both methods have some common solutions shown in Table 1.

In addition to these solutions, we gain forty-three new TWSs $u_1(\xi), u_2(\xi), u_5(\xi)−u_7(\xi)$ and $u_9(\xi)−u_{46}(\xi)$ in this article that are not mentioned in Al-Ghafri et. al [51].

In addition, Zafar et. al [53] have also explored twenty solitary wave solutions from the conformable time-fractional (3+1)-dimensional *mKdV−ZK* equation through three integration schemes included as $\exp(\xi)$ function scheme, hyperbolic function scheme, and modified kudryashov scheme. As opposed to generate many wave solutions from the stated equation by the mentioned method in this research. Both approaches share some potential solutions, which are compared and contrasted in Table 2.

Apart from these solutions, further new forty-four new exact TWSs $u_1(\xi)−u_8(\xi), u_{10}(\xi)$ and $u_{12}(\xi)−u_{46}(\xi)$ are established in this article that are not mentioned in Zafar et. al [53].

## 5.Conclusion

Through the use of the new auxiliary equation approach and the unified technique, we have achieved the exact and precise travelling wave soliton solutions for the (3+1)-dimensional *mKdV−Zk* model. Under certain scenarios, the travelling wave solutions can be expressed as

**Table 1. Comparison between attained solutions with Al-Ghafri et. al [51] solutions.**

| Obtained solutions | Al-Ghafri et. al [51] solutions |
|---|---|
| Taking $p = q = r = l = 1,\ m = 2, n = 3, \beta = -2,$ and $u_3(x,y,z,t) = F$, then the solution becomes $F = -\frac{3}{2}tanh\left(\frac{1}{2}\left(x + y + z + \frac{3}{2}t\right)\right).$ | In Eq (44) $u(x, y, z, t) = \frac{\varepsilon}{2}\sqrt{-\frac{6c_2(k_1^2+k_2^2+k_3^2)}{\delta}}tanh\left(\frac{\sqrt{c_2}\xi}{2}\right).$ Taking $k_1 = k_2 = k_3 = c_2 = \alpha = 1,\ \varepsilon = -1,\ \delta = -2$ and $u(x,y,z,t) = F$, then the solution becomes $F = -\frac{3}{2}tanh\left(\frac{1}{2}\left(x + y + z + \frac{3}{2}t\right)\right).$ |
| Taking $p = q = r = l = 1, m = 2, n = 3, \beta = -2,$ and $u_4(x, y, z, t) = F$ then the solution becomes $F = -\frac{3}{2}coth\left(\frac{1}{2}\left(x + y + z + \frac{3}{2}t\right)\right).$ | In Eq (45) $u(x, y, z, t) = \frac{\varepsilon}{2}\sqrt{-\frac{6c_2(k_1^2+k_2^2+k_3^2)}{\delta}}coth\left(\frac{\sqrt{c_2}\xi}{2}\right).$ Taking $k_1 = k_2 = k_3 = c_2 = \alpha = 1,\ \varepsilon = -1,\ \delta = -2$ and $u(x,y,z,t) = F$, then the solution becomes $F = -\frac{3}{2}coth\left(\frac{1}{2}\left(x + y + z + \frac{3}{2}t\right)\right).$ |
| Taking $p = q = r = m = 1, l = -1, n = 2, \beta = -2$ and $u_8(x,y,z,t) = F$, then the solution becomes $F = -3\sqrt{2}coth(\sqrt{2}(x + y + z + 12t)).$ | In Eq (89) $u(x, y, z, t) = 2\varepsilon\sqrt{\frac{3c_2(k_1^2+k_2^2+k_3^2)}{\delta}}coth\left(\sqrt{-2c_2}\xi\right)$ Taking $k_1 = k_2 = k_3 = \alpha = 1, c_2 = -1,\ \delta = -18, \varepsilon = -3$ and $u(x,y,z,t) = F$, then the solution becomes $F = -3\sqrt{2}coth(\sqrt{2}(x + y + z + 12t)).$ |

**Table 2. Comparison between attained solutions with Zafar et. al [53] solutions.**

| Obtained solutions | Zafar et. al [53] solutions |
|---|---|
| Taking $p = q = r = m = l = 1, n = \sqrt{3}, \beta - 2$ and $u_9(x, y, z, t) = F$, then the solution becomes $F = \frac{3}{2} tan\left(\frac{1}{2}\left(x + y + z - \frac{3}{2}t\right)\right).$ | In Eq (3.11) $u_9(\epsilon) = \frac{\sqrt{6}\sqrt{-ek^2 - fp^2 - gq^2}}{2\sqrt{d}}$ $tan\left(\frac{kx + py + qz - \frac{p^3}{\lambda}}{2}\right).$ Taking $k = p = q = \lambda = e = f = g = 1, d = -2$ and $u(x,y,z,t) = F$ then the solution becomes $F = \frac{3}{2} tan\left(\frac{1}{2}\left(x + y + z - \frac{3}{2}t\right)\right).$ |
| Taking $p = q = r = l = m = 1, n = \sqrt{5}, \beta = -2$ and $u_{11}(x,y,z,t) = F$, then the solution becomes $F = -\frac{3}{2} tanh\left(\frac{1}{2}\left(x + y + z + \frac{3}{2}t\right)\right).$ | In Eq (3.7) $u_1(x,y,z,t) = \frac{-\sqrt{6}\sqrt{-ek^2 - fp^2 - gq^2}}{2\sqrt{d}}$ $tanh\left(\frac{kx + py + qz - \frac{1}{2}\left(-ek^3 - fkp^2 - gkq^2\right)\frac{t^\lambda}{\lambda}}{2}\right).$ Taking $k = p = q = \lambda = e = f = g = 1, d = -2$ and $u(x,y,z,t) = F$, then the solution becomes $F = -\frac{3}{2} tanh\left(\frac{1}{2}\left(x + y + z + \frac{3}{2}t\right)\right).$ |

rational, hyperbolic, and trigonometric functions. However, these techniques produce abundantly distinct free parametric values that depict geometrically kink-shaped soliton solutions, anti-kink-shaped solutions, bell-shaped soliton solutions, and periodic solutions. In addition, the reactions of various nonlinearities strengths, wave velocity, and other model factors are investigated. Several of the solutions found are completely fresh and have not been discussed in any of the previous research. To elucidate the attained outcomes, we have looked at surface, contour, and combined 2-D diagrams. For distinct numerical values of parameters, we demonstrated several 2-D graphs, and watching the graph makes it easy to observe the wave velocity. These extensive results can be helpful resources for investigators as they look at the geometrical structure and understand the system's physical interpretation. The outcomes of other research studies that are currently available are compared with the findings of this work. The calculation is straightforward, yields more unique results than other techniques currently used, and has a wider range of applications due to decreased consistency and computational tasks. However, these methods fail to produce exact solutions for some fractional balance number models. To summarize, both the unified scheme and the NAE scheme are effective, compatible, and simple approaches for obtaining full wave solutions with a variety of free parameters, providing significant insights into wave profiles across varied contexts. These techniques yielded precise travelling wave soliton solutions, and they are strongly recommended for future research on nonlinear models, which hold significance in mathematical physics.

## Acknowledgments

The authors would like to thank the editor of the journal and anonymous reviewers for their generous time in providing detailed comments and suggestions that helped us to improve the paper.

## Author Contributions

**Conceptualization:** S. M. Yiasir Arafat.

**Data curation:** S. M. Yiasir Arafat, M. M. Rahman.

**Formal analysis:** S. M. Yiasir Arafat.

**Investigation:** M. Asif, M. M. Rahman.

**Methodology:** S. M. Yiasir Arafat, M. Asif.

**Software:** S. M. Yiasir Arafat.

**Supervision:** M. M. Rahman.

**Writing – original draft:** S. M. Yiasir Arafat.

**Writing – review & editing:** S. M. Yiasir Arafat, M. Asif.

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
