## [Decision Letter · Decision Letter 0]

16 Jul 2024

PONE-D-24-25524Nonlinear dynamic wave properties of travelling wave solutions in ion-acoustic wavesPLOS ONE

Dear Dr. Arafat,

Thank you for submitting your manuscript to PLOS ONE. After careful consideration, we feel that it has merit but does not fully meet PLOS ONE’s publication criteria as it currently stands. Therefore, we invite you to submit a revised version of the manuscript that addresses the points raised during the review process.

We look forward to receiving your revised manuscript.

Kind regards,

Ghulam Rasool

Academic Editor

PLOS ONE

Journal Requirements:

Reviewers' comments:

Reviewer's Responses to Questions

**Comments to the Author**

1. Is the manuscript technically sound, and do the data support the conclusions?

Reviewer #1: Yes

Reviewer #2: Partly

Reviewer #3: Partly

2. Has the statistical analysis been performed appropriately and rigorously? 

Reviewer #1: Yes

Reviewer #2: N/A

Reviewer #3: N/A

3. Have the authors made all data underlying the findings in their manuscript fully available?

Reviewer #1: No

Reviewer #2: Yes

Reviewer #3: Yes

4. Is the manuscript presented in an intelligible fashion and written in standard English?

Reviewer #1: No

Reviewer #2: No

Reviewer #3: No

5. Review Comments to the Author

Reviewer #1: 1. There are many grammatical errors. Kindly check them. (Full stop and commas)

2. What is the difference between solution sets 1 and 2 on page 12. They only differ in signs.

3. The references are not in a uniform style.

4. What are the applications of the obtained solutions?

5. What are the advantages and disadvantages of the proposed methods?

6. The introduction section is lacking drastically with the introduction to the Solitons. The authors are requested to read the following latest articles to enhance the introduction section and cite them properly.

doi.org/10.1155/2014/601961

doi.org/10.3390/sym15020360

doi: org/10.1016/j.asej.2022.101883

doi.org/10.3390/sym14122574

doi.org/10.3390/math10183400

Reviewer #2: Dear Authors,

Please note that the report of your paper is attached to this submission. Please have a look at each comment and revise it.

Your paper needs more and more corrections.

Best wishes

Reviewer

Reviewer #3: I read this paper in detail and the idea of this paper is interesting. However, I suggest the following issues should be revised and modified before it can be considered for publication. My comments are as follows:

1. The abstract should be improved. The author should discuss the significance and advantages of this study according to the title of the manuscript.

2. The physical significance of the model and the phenomenon should be described in the manuscript.

3. A comparative study should be included. What is your innovative work? The authors should add a section and compare your results with others authors results in literature.

4. What are previous studies on the governing model?

5. Please confirm the punctuation mark for each equation, please check the all equations to make sure they are correct and also dot or commas are missing in many equations.

6. The Physical explanation of graphical results are not sufficient. Furthermore, relevant applications should be added to make this article to attract the attention of multidisciplinary audience.

7. What is the reason behind choosing this method though many other methods are also present in the literature? The authors are requested to highlight the same in the section "Introduction".

8. In conclusion section, mention few shortcomings of the proposed method and this should also be expanded to include more comprehensive discussion.

9. For better presentation of the paper, the authors should cite and discuss recent works on the solitons in this field:

• On the Gaussian traveling wave solution to a special kind of Schrödinger equation with logarithmic nonlinearity. Modern Physics Letters B, 36(02), 2150543. doi: 10.1142/S0217984921505436

• Exploration conversations laws, different rational solitons and vibrant type breather wave solutions of the modify unstable nonlinear Schrödinger equation with stability and its multidisciplinary applications, Optical and Quantum Electronics 56 (2024) 420.

• Linear structure and soliton molecules of Sharma-Tasso-Olver-Burgers equation. Physics Letters A, 452, 128430. doi: https://doi.org/10.1016/j.physleta.2022.128430

• Exploring fractional-order new coupled Korteweg-de Vries system via improved Adomian decomposition method, Plos one, 19 (5) (2024), e0303426.

• Analytical optical solutions to the nonlinear Zakharov system via logarithmic transformation. Results in Physics, 56, 107298. doi: https://doi.org/10.1016/j.rinp.2023.107298

This paper has some technical issues as mentioned above; Therefore, I recommend for publication after major revision in Plos One.

6. PLOS authors have the option to publish the peer review history of their article (what does this mean?). If published, this will include your full peer review and any attached files.

Reviewer #1: No

Reviewer #2: No

Reviewer #3: No

---

## [Author Response · Author response to Decision Letter 0]

24 Aug 2024

Dear Editor, we are Bangladeshi authors and we have no funds for this research. For some of my technical faults, the submission process showed APC. Please see my cover letter and help me this situation.

---

## [Decision Letter · Decision Letter 1]

3 Sep 2024

Nonlinear dynamic wave properties of travelling wave solutions in  in (3+1)-dimensional mKdV-ZK model

PONE-D-24-25524R1

Dear Dr. Arafat,

We’re pleased to inform you that your manuscript has been judged scientifically suitable for publication and will be formally accepted for publication once it meets all outstanding technical requirements.

Kind regards,

Ghulam Rasool

Academic Editor

PLOS ONE

Additional Editor Comments (optional):

Reviewers' comments:

Reviewer's Responses to Questions

**Comments to the Author**

1. If the authors have adequately addressed your comments raised in a previous round of review and you feel that this manuscript is now acceptable for publication, you may indicate that here to bypass the “Comments to the Author” section, enter your conflict of interest statement in the “Confidential to Editor” section, and submit your "Accept" recommendation.

Reviewer #1: All comments have been addressed

Reviewer #2: All comments have been addressed

Reviewer #3: (No Response)

2. Is the manuscript technically sound, and do the data support the conclusions?

Reviewer #1: Yes

Reviewer #2: Yes

Reviewer #3: Yes

3. Has the statistical analysis been performed appropriately and rigorously? 

Reviewer #1: Yes

Reviewer #2: N/A

Reviewer #3: N/A

4. Have the authors made all data underlying the findings in their manuscript fully available?

Reviewer #1: Yes

Reviewer #2: Yes

Reviewer #3: Yes

5. Is the manuscript presented in an intelligible fashion and written in standard English?

Reviewer #1: Yes

Reviewer #2: Yes

Reviewer #3: Yes

6. Review Comments to the Author

Reviewer #1: The authors have commented all the points raised by the reviewer. My recommendation is, to accept the manuscript in the present form.

Reviewer #2: Dear Authors,

Thank you so much for addressing all the provided comments.

The paper is now fine.

Best wishes

Reviewer

Reviewer #3: I read this revised version of paper in detail. The authors have done an excellent job addressing most of the queries. However, I suggest the following issue should be revised before it can be considered for publication. My comments are as follows:

In subsection 2.2 at page 5, the travelling wave solutions in Case 1 and Case 2 are single solutions or composite solutions. The authors should correct it by using dot or commas.

After equation (3.4) at page 6, where k_0 and k_1are constants real or complex? If it is real then the authors should mention conditions on parameters in solution sets 1 and 2 on k_0 and k_1 obtained solutions.

There are a few typos and grammatical errors like dot or commas are missing in some equations. It is strongly recommended that you proofread thoroughly.

For improvement in the introduction and in order to explain the future direction on the below latest work is cited:

* https://doi.org/10.1016/j.rinp.2024.107601

* https://iopscience.iop.org/article/10.1088/1402-4896/ad6d18

* https://doi.org/10.1016/j.rinp.2024.107431

A minor revision to enhance these sections are suggested.

7. PLOS authors have the option to publish the peer review history of their article (what does this mean?). If published, this will include your full peer review and any attached files.

Reviewer #1: No

Reviewer #2: No

Reviewer #3: No

---

## [Editor Report · Acceptance letter]

19 Sep 2024

PONE-D-24-25524R1 

PLOS ONE

Dear Dr. Arafat, 

I'm pleased to inform you that your manuscript has been deemed suitable for publication in PLOS ONE. Congratulations! Your manuscript is now being handed over to our production team.

Kind regards, 

on behalf of

Dr. Ghulam Rasool 

Academic Editor

PLOS ONE